# Feasibility of Leukemia-Derived Exosome Enrichment and Co-isolated dsDNA Sequencing in Acute Myeloid Leukemia Patients: A Proof of Concept for New Leukemia Biomarkers Detection

**DOI:** 10.3390/cancers14184504

**Published:** 2022-09-16

**Authors:** Simona Bernardi, Mirko Farina, Katia Bosio, Anna Di Lucanardo, Alessandro Leoni, Federica Re, Nicola Polverelli, Alessandro Turra, Enrico Morello, Eugenia Accorsi Buttini, Tatiana Zollner, Cristian Bonvicini, Michele Malagola, Domenico Russo

**Affiliations:** 1Department of Clinical and Experimental Sciences, University of Brescia, Bone Marrow Transplant Unit, ASST Spedali Civili, 25123 Brescia, Italy; 2Centro di Ricerca Emato-Oncologica AIL (CREA), ASST Spedali Civili, 25123 Brescia, Italy; 3Molecular Markers Laboratory, IRCCS Istituto Centro San Giovanni Di Dio Fatebenefratelli, 25123 Brescia, Italy

**Keywords:** exosomes, extracellular vesicles, cell of origin enrichment, acute myeloid leukemia, next-generation sequencing, exosomal DNA, liquid biopsy

## Abstract

**Simple Summary:**

The present pilot study aimed at investigating the feasibility of a leukemia-derived exosome enrichment approach followed by exosomal dsDNA target re-sequencing for adult Acute Myeloid Leukemias (AML) marker detection. To our knowledge, this is the first time that a proof-of-concept combining a leukemia-derived exosome enrichment strategy based on a commercial CE-IVD kit and next-generation sequencing was applied in a cohort of adult AML patients. The reported approach is easy, quick and user friendly and gives the possibility of obtaining a good quantity of exosomal dsDNA (composed of exosomal cargo and surrounding DNA) suitable for further analysis. The time-effective procedure opens up future effective clinical applications. This pilot study presents the potential of a proof-of-concept based on exosome analysis to be applied in clinical practice, as well as the feasibility of this kind of investigations using a certified kit, avoiding many additional analyses. It may encourage further studies regarding extracellular vesicles in myeloid neoplasia.

**Abstract:**

Exosomes are extracellular vesicles playing a pivotal role in the intercellular communication. They shuttle different cargoes, including nucleic acids from their cell of origin. For this reason, they have been studied as carriers of tumor markers in different liquid biopsy approaches, in particular for solid tumors. Few data are available concerning exosomes as markers of myeloid neoplasia. To better understand their real potential and the best approach to investigate leukemic exosomes, we present the results of a pilot feasibility study evaluating the application of next-generation sequencing analysis of dsDNA derived from exosomes isolated in 14 adult patients affected by acute myeloid leukemias. In particular, leukemia-derived exosome fractions have been analyzed. The concentration of dsDNA co-extracted with exosomes and the number and types of mutations detected were considered and compared with ones identified in the Bone Marrow (BM) and Peripheral Blood (PB) cells. Exosomal DNA concentration, both considering the cargo and the DNA surrounding the lipid membrane resulted in a linear correlation with leukemic burden. Moreover, exosomal DNA mutation status presented 86.5% of homology with BM and 75% with PB. The results of this pilot study confirmed the feasibility of a leukemia-derived exosome enrichment approach followed by exosomal dsDNA NGS analysis for AML biomarker detection. These data point to the use of liquid biopsy in myeloid neoplasia for the detection of active leukemic cells resident in the BM via a painless procedure.

## 1. Introduction

### 1.1. Extracellular Vesicles (EVs) and Liquid Biopsy

Extracellular Vesicles (EVs) are membrane vesicles of 30–1000 nm in size and are essential mediators of cell-to-cell communication [1]. They shuttle a variety of cargoes, from nucleic acids to proteins, and may be isolated both in in vitro models and in many body fluids in vivo [2]. Up to now, no consensus has yet been achieved about EVs classification, even if the International Society for EVs (ISEV) has recently grouped these particles into small EVs and large EVs by size, giving also additional criteria, such as their origin. Among EVs, exosomes are endosomal-derived small-vesicles, presenting a median diameter about ~100 nm [3]. The expression of peculiar markers, such as the tetraspanins CD9, CD63 and CD81, distinguishes exosomes from other EVs. The capability of EVs to carry markers of the cell of origin has made them one of the main actors of the liquid biopsy, a recently developed approach based on the detection of circulating cell markers. Liquid biopsy has been widely explored in oncology and only lately has it been introduced in hematology, with a special focus on lymphoid malignancies [4,5]. Despite the presence of some in vitro data about the characterization of exosomes derived from myeloid leukemic cells, few studies have reported their investigation in patients affected by myeloid neoplasia [6].

### 1.2. Acute Myeloid Leukemias (AMLs)

Acute Myeloid Leukemias (AMLs) are ones of the most challenging myeloid disorders. AML are clonal disorders of the hematopoietic stem cells committed to the myeloid lineage that most frequently occur in older people (60–65 years), and their incidence increases progressively with age [7,8]. AMLs are characterized by a blocked or severe impaired differentiation of hematopoietic cells, resulting in a progressive accumulation of pathological cells (blasts) in various stages of incomplete maturation within the bone marrow (BM). This causes various levels of cytopenia in the peripheral blood (PB). In the absence of therapy, AML leads to death in a period of time ranging from a few days to a few months. AML can arise as a “de novo” form or after a previous hematopoietic disorder, such as Myelodysplastic Syndromes (MDS). AML includes a group of genetically heterogeneous diseases characterized by somatic acquired mutations [8,9,10,11], and germline predisposition [12,13,14] for both AML and MDS [15] was recognized and introduced in the latest revised World Health Organization (WHO) classification of hematological malignancies [16]. Immunophenotypic, cytogenetic and molecular analyses are conventionally used for AML characterization and classification, which are mainly based on the presence/absence of specific chromosomal rearrangements and fusion genes. AML characterization is fundamental in driving the therapeutic strategy [17,18] and monitoring the Minimal Residual Disease (MRD) [19,20,21]. For a long time, the knowledge of the molecular heterogeneity of AML only allowed a better understanding of the disease pathogenesis or an improved risk stratification of AML patients. Indeed, for decades the treatment options for AML patients were limited to cytotoxic chemotherapeutic without targeted therapies, and allogeneic Hematopoietic Stem Cells Transplantation (allo-SCT). Recently, as a result of the increasing knowledge of the AML biology and genomic landscape, multiple new therapies have been approved, allowing for an increasingly targeted personalized medicine [22,23]. Although the majority of patients have morphologically complete remission after treatment with intensive chemotherapy, the relapse rate remains high. Indeed, almost 50% of AML patients relapse and die from refractory disease after an initial response to leukemia therapy [24]. Overall, AMLs remain a significant challenge for hematologists and further studies are required to investigate the prognostic and predictive value of surrogate biomarkers. These will allow the identification of new reliable therapeutic and disease markers. Additional biological insights and the application of new technologies are encouraged in order to overcome these troubles. Could exosomes be reliable new tools to partially overcome these limitations?

### 1.3. A proof of Concept for the Study of AML Markers by EVs

In the present pilot study, we investigated the feasibility of a leukemia-derived exosomes enrichment approach in adult AML patients at different disease stages. The exosomal enrichment and the contemporary dsDNA extraction have been performed by a commercial kit certified both for the endosomal derivation of the enriched EVs (exosomes) and for diagnostic application. The present pilot study combined leukemic exosome enrichment with exosomal dsDNA Next-Generation Sequencing (NGS) analysis [25]. In order to test the feasibility and reliability of the proposed approach, we evaluated the quantity of exosomal dsDNA as a potential biomarker of AML. In the present study, the total isolated DNA has been considered (hereinafter “exosomal DNA”), both shuttled as cargo of the vesicles and as cfDNA surrounding the lipid membrane. Secondly, we compared the results obtained analyzing the exosomal dsDNA with what was observed in the cells. Moreover, we evaluated the different types of molecular leukemic markers detectable in the studied exosome fraction to test the absence of bias of selection of the dsDNA molecules during the vesicles’ biogenesis, as well as performed a DNA analysis.

## 2. Materials and Methods

### 2.1. Patients

Fourteen patients affected by AML at different disease statuses afferent to the Blood Disease and Stem Cells Transplantation Unit at ASST Spedali Civili of Brescia (Italy) were enrolled in the clinical trial NP 4344, a monocentric non-pharmacological pilot clinical trial approved by the Ethics Committee of Province of Brescia (Italy) in 2020. One healthy 46 year-old male, one healthy 50 year-old female, and one healthy 53 year-old female without familial history for hematological malignancies served as healthy controls. The study was conducted according to the guidelines of the Declaration of Helsinki and all the enrolled subjects gave their written informed consent. The main characteristics of the patients are reported in Table 1. Briefly, 7 male and 7 female AML patients were enrolled, with a median age at enrollment of 48 years (range 30–77 years). Patients were evaluated following the International guidelines and clinical practice [7,26].

A total of 101 samples were analyzed (31 BM samples; 34 PB samples, and 34 leukemia-derived exosomes). No BM specimens were sampled from healthy controls (Specimen 32, 33, and 34) because of ethical consideration. Patients’ disease statuses at the moment of each specimen are summarized in Table 1. BM sampling was performed following the conventional clinical practice and no additional aspirate was required.

### 2.2. Blood Sampling and Plasma and Cells Isolation

At each time point, 10 mL of PB and BM were sampled in EDTA tubes (5 mL for each tube) for all enrolled patients. Only PB samples were obtained by the healthy controls. A total of 101 samples were collected and analyzed. Fresh PB and BM samples were treated within 4 h from sampling.

PB samples were immediately centrifuged at 2800 rpm for 15′ to remove the supernatant containing the plasma. The plasma was stored. The buffy coat layer, with some red blood cells, was then removed and transferred to a new 15 mL tube. A lysing buffer solution was prepared (NH 4 Cl 0.155 mM, EDTA 0.125 mM, NaHCO 3 12 mM, stored at 4 °C) for the lysis of red blood cells. The procedure was grouped inro two lysis steps: 10′ at room temperature (RT) followed by a centrifuge at 2800 rpm for 10′. A second lysis was performed at RT for 5′ followed by a centrifuge at 2800 rpm for 5′. Then, two washing steps in PBS buffer (centrifuge at 2000 rpm for 10′) concluded the procedure. 

BM samples were treated with Ficoll–Paque density gradient centrifugation. Briefly, the collected BM samples, supplemented with anticoagulants and EDTA, were diluted 1:1 with PBS buffer. Then, they were carefully laired onto the required Ficoll–Paque volume. A centrifugation step was performed at 2000 rpm for 30′ without braking. The upper layer was discarded and the mononuclear cell layer was transferred to a new 15 mL centrifuge tube. Finally, two washing steps were performed in PBS buffer (centrifuge at 2000 rpm for 10′). After the cell count, the dried cell pellets were stored at −80 °C. 

The isolated cells and plasma were stored at −80 °C and −20 °C upon analysis, respectively.

### 2.3. Cellular DNA Extraction

Cellular genomic DNA was manually extracted by a commercial kit QIAamp DNA Mini kit (Qiagen, Hilden, Germany), according to the manufacturer’s Instructions. The input content of the cells was overall 5 × 10^6^ cells. Firstly, the cells were resuspended in 200 µL of PBS and 20 µL of Qiagen Protease and 200 µL Buffer AL was added. Samples were pulsed, vortexed and incubated at 56 °C for 10′. Then, 200 µL of 99% ethanol was added and the mixes were transferred to the QIAmp Mini spin column and centrifuged at 8000 rpm for 1′. Secondly, two washes were performed in Buffer AW1 (centrifuge at 8000 rpm for 1′) and Buffer AW2 (centrifuge at 14,000 rpm for 3′). After an incubation with 50 µL Buffer AE and centrifugation at 8000 rpm for 1′, the purified DNA was finally eluted and stored at −20 °C.

### 2.4. Exosomal dsDNA Isolation 

In order to select the best isolation kit for our purpose, 4 samples were treated in parallel with a commercial kit for total plasmatic exosomes isolation (Invitrogen) and with a commercial kit providing an enrichment in exosomes, derived from malignant cells (Exosomics Siena S.p.A, Siena, Italy).

Total exosomes’ isolation was performed by Total Exosome Isolation reagent (Invitrogen) following the manufacturer’s instruction. Briefly, 1 mL of plasma was centrifuged at 2000× *g* for 30′ to eliminate any cellular debris. The isolation of exosomes was accomplished with proteinase treatment in order to achieve a better purification. The exosomes’ pellets were resuspended in 200 µL PBS buffer and stored at 4 °C overnight. Then, the extraction of exosomal dsDNA was assessed according to QIAamp DNA Mini kit (Qiagen, Hilden, Germany) protocol starting from 200 µL of exosomes resuspended in 200 µL of PBS, as described above for cellular DNA extraction. 

The exosome fraction enriched for vesicles derived from malignant cells was isolated from 1 mL of plasma by the SeleCTEV™ Low Input DNA Enrichment kit (Exosomics Siena SpA, Siena, Italy). The SeleCTEV™ Low Input DNA Enrichment kit is CE-IVD certified and all the tests on the type and quality of enriched EVs and on exosomal dsDNA are declared by the manufacturer. The kit allows for direct exosomal dsDNA (exoDNA) extraction and the isolation is based on an affinity method, taking advantage of a proprietary peptide [27]. Briefly, 1 mL of plasma was diluted with 1 mL of 1 × isolation buffer. Then, 2 µL of protease inhibitor cocktail (Sigma), in order to prevent degradation, and 20 µL of isolation agent were added. The reaction mix was incubated at room temperature (RT) for 2 h. The samples were then centrifuged 15′ at 16,000× *g* at RT. The supernatant containing the reagents was discharged and 1 mL of 1 × isolation buffer was added to the pellet and subsequently spinned at 7000 g for 7′ at RT twice. Proteinase K and lysis buffer were added in 1:10 ratio. The samples were incubated at 56 °C for 1 h. Then, 200 µL of 99% ethanol was added to each sample, the mixture was transferred on a new column and then was centrifuged at 10,000× *g* for 1′. The samples were then washed with different washing buffers, following the manufacturer’s instructions. The exosomal dsDNA was evaluated for quality and quantity and stored at −20 °C until analysis.

After the dsDNA quality control was performed based on quantification of the 230/260 and 260/280 ratio, the SeleCTEV™ Enrichment kit was used and the experiment described hereinafter refers to exosomal dsDNA extracted by enrichment. The exosomal dsDNA quality control results are reported in Appendix A.

Exosomal dsDNA was quantified by the fluorometric method using the Quant-iT PicoGreen dsDNA Assay kit (Thermo Fisher Scientifics, Waltham, MA, USA).

### 2.5. Exosomal dsDNA Amplification by Whole Genome Amplification (WGA)

Exosomal dsDNA was amplified by the REPLI-g^®^ Single Cell Kit following the standard protocol provided by the manufacturer. This kit was chosen based on the previously reported efficiency and suitability for further NGS investigation [28]. Briefly, 15  µL of exosomal dsDNA and 2 µL of denaturation buffer DLB were incubated at RT for 3′. Later, 3  µL of Stop Solution was added. A WGA master mix containing 29 μL REPLI-g sc reaction buffer and 2 μL of REPLI-g sc DNA polymerase was added to the mix followed by amplification at 30  °C for 8 h and inactivation at 65  °C for 3′. The obtained amplified exosomal dsDNA samples were labeled as Amp-exDNA.

The Amp-ex DNA was loaded onto a 1% E-Gel Sybr-safe electrophoresis system (Thermo Fisher Scientifics, Waltham, MA, USA) for quality control.

### 2.6. Next Generation Sequencing (NGS) Analysis

Cellular and Amp-exDNA was analyzed by NGS using a custom panel for target resequencing based on Nimblegen technology (Roche Diagnostics Italia, Monza, Italy). The custom panel included the coding and regulatory sequences of the following genes, genes known to be involved in myeloid leukemogenesis and suggested to be investigated by the recent international guidelines and by experts in the field: *ASXL1, BCOR, NRAS, TP53, RUNX1, CEBPA, FLT3, EZH2, IDH1, IDH2, NPM1, DNMT3A, TET2, CBL, KRAS, ETV6, SF3B1, SRSF2, U2AF1, ZRSR2, GATA2, TERT, TERC, SRP72*, and *ANKRD26*. The protocol for library generation was performed following the manufacturer’s instructions, as previously reported, with 500ng of DNA as an input [25]. DNA libraries were sequenced by the MiniSeq and MiSeq Illumina NGS platform (Illumina, San Diego, CA, USA) using 2 × 150 sequencing (Highoutput and V3 for 600 cycles sequencing kit, respectively).

A total of 5 NGS sequencing runs were performed and bioinformatic analysis was conducted on .vcf files automatically generated by the MiniSeq instrument by wAnnovar (Wang Genomics) [29]. Different databases, such as dbSNPs, ClinVar, and COSMIC, were interrogated and the variant frequency in the population was considered (i.e., ExAC and 1000 Genome). A coverage at 1000× was considered as acceptable for the consistence of the results [30]. 

Novel mutations have been confirmed by a second target resequencing analysis by Sophia Genetic Myeloid Solution (Sophia Genetics, Saint Sulpice, Switzerland) following the manufacturers’ instruction. DNA libraries were sequenced by the MiSeq Illumina NGS platform with 2 × 150 sequencing by V3 for 600 cycles. Bioinformatic analysis on .Fastq files was conducted by Sophia DDM, an artificial intelligence-, machine learning-, and cloud-based software. It automatically queries a wide ranges of databases and performs an algorithm-supported variant classification with OncoPortal™. Sophia Genetic Myeloid Solution with bioinformatic analysis by Sophia DDM is a CE-IVD-certified diagnostic workflow for AML.

### 2.7. Statistical Analysis

Standard descriptive statistics were used to summarize the samples’ characteristics [31]. Categorical variables were compared using the Chi-squared test. Simple linear regression was used to estimate the relationship between exosomal DNA concentration and the leukemia burden. Statistical analyses were performed with Prism (GraphPad Version 7.0).

## 3. Results

### 3.1. Correlation between Exosomal dsDNA and Leukemia Burden

We firstly evaluated the correlation between exosomal dsDNA concentration and the leukemia burden. A linear regression (R = 0.849) was observed considering the quantification performed by Quant-iT PicoGreen on exosomal dsDNA extracted by the peptide affinity method and the leukemia burden assessed by routine cytofluorimetric analysis, considering cells presenting with an aberrant immunophenotype (Figure 1). The cytofluorimetric analysis conducted on all the sampled cells included the detection of the following antigens: CD45, CD2, CD22, CD14, CD8, CD34, CD5, CD20, CD13, CD4, CD7, CD10, CD15, HLA-DR, CD33, CD19, CD117, CD56, CD3, CD11b, CD64, and CD16.

This evidence has suggested an increased release of exosomes related with the increment of leukemic blasts.

### 3.2. Analysis of Mutations Detected in Cellular and Exosomal dsDNA

NGS analysis allowed for the detection of variants in all sequenced samples. Variants known as polymorphisms were excluded from analysis. In healthy controls’ samples, only variants classified as polymorphisms were detected.

We compared the mutations detected in PB and BM cells with those identified in exosomal dsDNA. The number of identified mutations in the different sequenced samples is reported in Table 2. No mutation resulted related with clonal hematopoiesis. At least one mutation has been detected on exosomal dsDNA in 6 out of 10 (60%) specimens sampled at the moment of Complete Remission (CR), while only 4/10 (40%) and 0/10 (0%) samples revealed mutations on Bone Marrow (BM) and PB cellular DNA, respectively.

Considering the detected non-unique mutations, the highest number of mutations was detected on exosomal dsDNA (69 non-unique mutations), while the lowest was in PB cellular DNA (44 non-unique mutations). When filtering for unique mutations, BM and exosomal dsDNA presented 86.5% of homology (Figure 2A), higher than the homology between PB and exosomal dsDNA (Figure 2B) and between BM and PB DNA (75% and 77%, respectively) (Figure 2C). The total number of detected mutations have been reported and rendered in Figure 2D. 

The types of mutations were also considered. The complete list of the detected mutations is reported in Table 3.

The variants were divided into Nonsynonymus Single Nucleotide Variants (SNV), Stopgain mutations and Frameshift mutations. Nonsynonymus SNV was the most represented type both in cells (65%) and in exosomes (67%), while Stopgain mutations were the less frequent ones (8% and 9% in cells and exosomes, respectively). The frequency of each mutation type is graphically represented in Figure 3.

We finally evaluated the types of genes mutated. Genes were divided into different categories based on their main function: signal transduction, epigenetic and transcription regulators, metabolism, splicing, polymerase activity, and other functions. Both in cells and in exosomes, the epigenetic and transcription regulators resulted to be the most mutated genes (50% and 60%, respectively). Conversely, genes involved in metabolism and with polymerase activity were the less mutated ones in both the sequenced materials (4% and 3% in cells and exosomes, respectively). The detailed frequency of mutated gene types is graphically represented in Figure 4.

No statistically significant difference resulted from comparing the frequency of the type of mutations or the function of the mutated genes.

## 4. Discussion

Extracellular vesicles, exosomes in particular, have been identified as components of liquid biopsy approaches for many diseases, both oncological and non-oncological. In oncohematology, exosomes have been mainly investigated in lymphoid malignancies, while the majority of the data about myeloid neoplasia are derived from limited in vitro studies. Little is known about real exosomes’ capability to serve as leukemic markers shuttling detectable mutated nucleic acids [6]. We previously reported the correlation between exosomal dsDNA concentrations and disease status in adult AMLs [25], and similar results were observed in pediatric AML patients [32]. 

In the present pilot phase of a feasibility study, we have taken advantage of the synergy between a leukemia-derived exosome enrichment strategy, based on a commercial kit previously described [27], and a sensitive technology such as NGS. We tested the possibility of taking advantage of the application of a user-friendly commercial kit certifying the selection of EVs with endosomal origin. The peptide-based enrichment of EVs derived from malicious cells is expected to really improve the sensitivity of the approach, as experienced by Taylor and colleagues [33], and result in an increased quality and quantity of exosomal dsDNA. Similar approaches were explored in solid tumors with impressive results [34]. The commercial kit used in the present pilot study is CE-IVD certified, so the manufacturer and the certifications guarantee the type and quality of the isolated EVs (exosomes) together with the origin of the extracted dsDNA. One of the main limitations of the kit is that it has been developed for diagnostic purposes, so there is no possibility to manipulate the intermediate phases or to change the protocol and no step for exosome recovery and analysis is present in the protocol. The user easily obtains dsDNA of a certified origin starting from a low amount of plasma, but has no opportunity to stop the protocol and to perform a characterization on the isolated EVs. Based on the final aims, it may also be considered an advantage because the certifications allow reducing the analysis and the time of the procedures. These pro and cons must be considered during the experimental design. In addition to leukemia-derived exosome enrichment, the further NGS-based analysis on EV nucleic acids cargoes and on cfDNA have been reported in many contexts and the robustness of the approach has been widely described [35,36,37]. 

The correlation between the concentration of leukemic originated exosomal dsDNA and leukemia burden was firstly evaluated. A linear regression was identified (R = 0.849), and this confirmed the preliminary results previously presented [25]. Similar results were observed in solid tumor patients [38] and in patients affected by lymphoid malignancies [39,40,41]. The increase in exosome content and, consequently, of the quantity of their cargo in the case of leukemia cells, and the converse decrease in the case of CR or response to therapy, have been discussed in the last years for AML patients as well [42]. 

The NGS analysis of exosomal and cellular DNA allowed for a high-resolution profiling of the genomic landscape of our patients and for the representation of the blast DNA within the shed exosomal compartment. The observed homology is similar with what reported in solid tumors [43] and in other hematological malignancies [32]. Very interesting is the higher homology between the exosomes and BM than the homology between the exosomes and PB. This result is probably due to the presence of active leukemic cells still presenting with homing patterns and maintaining residence in the BM. Conversely, the reduced homology between mutations detected on exosomal dsDNA and PB cells may be affected by the study of exosomes instead of EVs at all [44] or by the detection of EVs released by non-circulating cells [45]. Exosomes may be released during particular moments of cell life and circulating leukemic cells could be less involved in exosomes release than BM leukemic cells, which are stimulated also by the BM niche [46,47]. So, exosomes may be considered to be effective messengers from the bone marrow. In some cases, the homology between exosomal DNA and tumor DNA is higher than what we observed. The difference may be due to the kit for exosomes’ isolation used in the present study, which limited the co-precipitation of cell-free DNA surrounding the vesicles [48]; cell-free DNA is known to carry many tumor mutations in oncologic patients [49]. In this study, we limited the analysis to the exosomes’ cargo and the DNA surrounding their membrane in order to better appreciate the real power of these small EVs, as similarly reported by other groups [50]. Differently, the reduced homology may be associated with the characteristics of AML as a “liquid” disease. This must be explored in a further enlarged study, since it was not the aim of the present study. Moreover, we noticed that different types of mutations and all the sequenced genes resulted in being well-represented in the isolated dsDNA. This aspect is particularly interesting, because it guarantees the investigators the capability to detect every type of alteration conventionally observed by DNA analysis. A previous study based on whole exome sequencing–NGS approach demonstrated that there was a robust representation of the tumor DNA within the exosomal compartment and that 95–99% of coding genome is present in these EVs [51]. Therefore, dsDNA cargo together with cfDNA surrounding the EVs may be confidently considered to be reliable markers because no genes and no mutations are selectively excluded during exosome biogenesis and during the process of analysis described in the present study. [52]. 

The EVs, as non-cellular copies of the cells of origin, have garnered clinical interest, and could be used when the cell of origin is neoplastic and it is not directly detectable. This is particularly evident when considering the samples, both BM and PB, collected at the moment of CR [53]. The mutation detectability on exosomal DNA at CR suggests the potential power of exosomes-based liquid biopsy for the detection of residual leukemic cells in AML and of early relapse, as observed in other malignancies [54]. This preliminary result, together with a consistent presence of leukemia-associated mutations at the different considered disease statuses and the negativity of healthy controls opens up the possibility of investigating circulating EVs as robust markers to confirm the response to treatment in adult AML patients, similarly to what is observed in pediatric AML cases [32,55]. 

## 5. Conclusions

In conclusion, these results reinforce what described both in vivo and in vitro for AML and other myeloid leukemias [56,57] about the capability of leukemic exosomes to carry molecular leukemic markers, both in their cargoes and through the surrounding DNA. Moreover, this pilot study of feasibility confirms the potentiality of liquid biopsy focused on exosomes, and small EVs at all, for a better monitoring of AML patients and to improve the early diagnosis of relapse after allo-SCT, in parallel with gold standard approaches [58,59]. 

The selection of subsets of patients better stratified by small EVs analysis will be interesting, together with the study of additional exosomal markers, such as transcripts. Further studies will also clarify the best timing of samples analysis. In the era of personalized and precise medicine [60], the combination of the available modern technologies and the scan of novel informative markers is mandatory in order to really improve AML patients’ management and to reduce the still-present challenges.

## Figures and Tables

**Figure 1 cancers-14-04504-f001:**
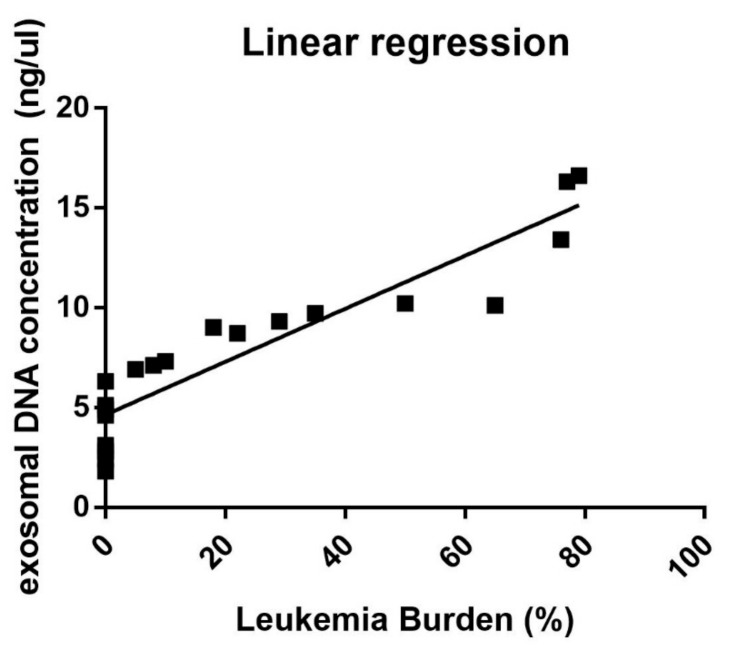
Linear regression analysis between leukemia burden estimated by cytofluorimetric analysis and exosomal DNA concentration expressed in ng/µL.

**Figure 2 cancers-14-04504-f002:**
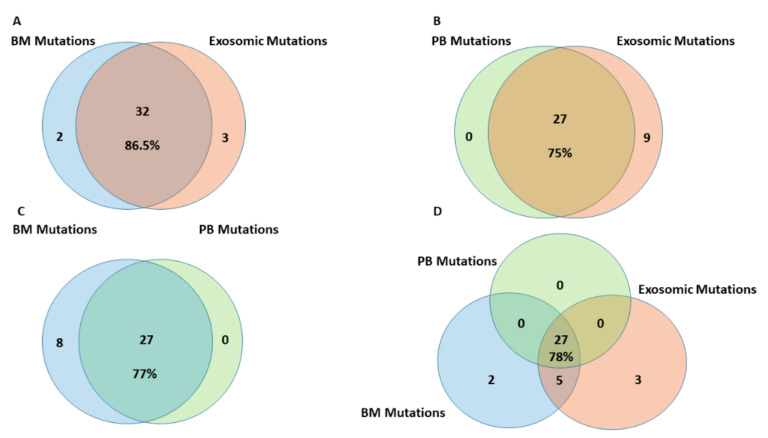
Homology of unique mutations identified in the different DNA sources. (**A**) Homology between BM cellular DNA and exosomal dsDNA; (**B**) Homology between PB cellular DNA and exosomal dsDNA; (**C**) Homology between BM and PB cellular DNA; (**D**) Homology between exosomal dsDNA, BM and PB cellular DNA. BM, PB and exosomal mutations (identified on the total dsDNA co-isolated with exosomes) are reported in blue, green and red diagrams, respectively.

**Figure 3 cancers-14-04504-f003:**
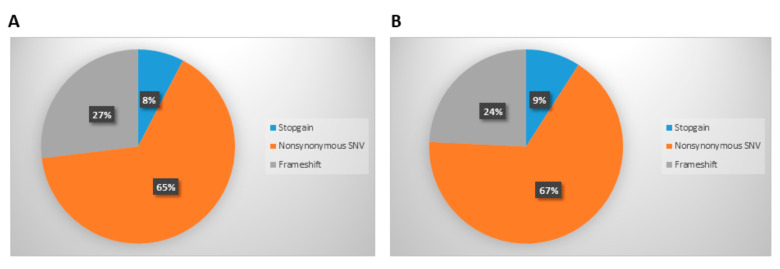
Graphical representation of the types of mutations detected in cells (**A**) and on dsDNA derived from exosome isolation (**B**).

**Figure 4 cancers-14-04504-f004:**
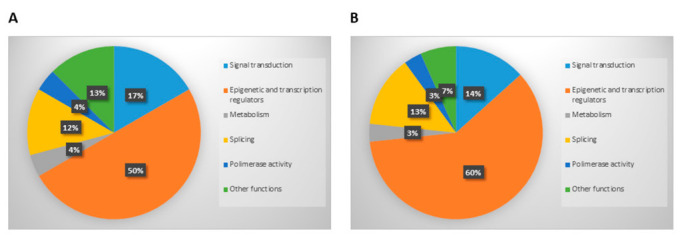
Graphic representation of the types of mutated genes, categorized on the bases of their biological main function, that were detected in cells (**A**) and on dsDNA derived from exosomes isolation (**B**).

**Table 1 cancers-14-04504-t001:** Clinical characteristics of the enrolled AML cases and the disease status at the moment of each specimen. Bone Marrow (BM), Peripheral Blood (PB) and leukemia-derived exosomes were analyzed for specimens 1–31. PB and leukemia-derived exosomes were analyzed for samples 32–34. * Mutations detected only at diagnosis.

*Case*	*Sex*	*Age*	*Disease Status at Enrollment*	*Known Mutations/Alteration*	*Disease Status at Sampling and Corresponding Specimens Number*
** *1* **	*M*	*61 y*	*Relapse post-alloSCT*	*WT1 overexpression*	*Relapse post-alloSCT (1)*
** *2* **	*F*	*47 y*	*Relapse*	*WT1 overexpression* *FLT3 D835Y* *NPM1 **	*Relapse (2)*
** *3* **	*F*	*30 y*	*CR post-alloSCT*	*WT1 overexpression* *DNMT3A*	*CR post-alloSCT (3)* *CR 3 months post-alloSCT (4)*
** *4* **	*M*	*71 y*	*Diagnosis of AML secondary to MDS*	*WT1 overexpression* *TP53*	*Diagnosis of AML secondary to MDS (5)*
** *5* **	*F*	*44 y*	*Relapse*	*WT1 overexpression*	*Relapse (6)* *Relapse post- second alloSCT (7)* *CR post-therapy Ven-Aza (8)*
** *6* **	*F*	*64 y*	*Relapse post-alloSCT*	*Complex Karyotype*	*Relapse post-alloSCT (9)*
** *7* **	*M*	*44 y*	*Relapse post-alloSCT*	*WT1 overexpression*	*Relapse post-alloSCT (10)* *CR post- therapy (11)*
** *8* **	*M*	*70 y*	*CR post relapse post-alloSCT*	*WT1 overexpression*	*CR post relapse post-alloSCT (12)*
** *9* **	*M*	*41 y*	*AML MRD+ pre-alloSCT*	*FLT3-ITD*	*AML MRD+ pre-alloSCT (13)* *CR post-alloSCT (14)*
** *10* **	*F*	*44 y*	*Pre-alloSCT*	*WT1 overexpression* *DNMT3A*IDH1*	*Pre-alloSCT (15)* *CR post-alloSCT (16)* *Relapse 7 months post-alloSCT (17)* *Relapse 10 months post-alloSCT (18)* *Post therapy Azacitidine (19)* *Post therapy Azacitidine (20)*
** *11* **	*M*	*77 y*	*Diagnosis of AML secondary to MDS*		*Diagnosis of AML secondary to MDS (21)* *CR post-alloSCT (22)* *Relapse post-alloSCT (23)*
** *12* **	*M*	*43 y*	*Pre-alloSCT*		*Pre-alloSCT (24)* *CR post-alloSCT (25)* *Relapse post-alloSCT (26)*
** *13* **	*F*	*49 y*	*Relapse 2* months *post-alloSCT*	*FLT3-ITD*	*Relapse 2 months post-alloSCT (27)* *Relapse 4 months post-alloSCT under target therapy (28)* *Relapse 6 months post-alloSCT under target therapy (29)*
** *14* **	*F*	*71 y*	*Pre-alloSCT*		*Pre-alloSCT (30)* *CR post-alloSCT (31)*

F = female; M = male; y = years old; alloSCT = allogeneic Stem Cells Transplantation; AML = Acute Myeloid Leukemia; MDS = Myelodysplastic Syndrome; CR = Complete Remission; MRD = Minimal Residual Disease; Ven-Aza = Venetoclax Azacytidine.

**Table 2 cancers-14-04504-t002:** Number of mutations detected in bone marrow cells’ DNA, peripheral blood cells’ DNA and circulating exosomes’ DNA in all the considered time points.

	*Material*	*BM Cells*	*PB Cells*	*Exo dsDNA*	*Disease Status*
*Specimen 1 (Case 1)*		*4*	*3*	*4*	*Relapse post-alloSCT*
*Specimen 2 (Case 2)*		*3*	*2*	3	*Relapse*
*Specimen 3 (Case 3)*		0	0	0	*CR post-alloSCT*
*Specimen 4 (Case 3)*		1	0	1	*CR 3 months post-alloSCT*
*Specimen 5 (Case 4)*		3	3	3	*Diagnosis of AML secondary to MDS*
*Specimen 6 (Case 5)*		3	3	4	*Relapse*
*Specimen 7 (Case 5)*		5	4	4	*Relapse post- second alloSCT*
*Specimen 8 (Case 5)*		*1*	*0*	*1*	*CR post-therapy Ven-Aza*
*Specimen 9 (Case 6)*		4	4	3	*Relapse post-alloSCT*
*Specimen 10 (Case 7)*		*3*	*2*	*3*	*Relapse post-alloSCT*
*Specimen 11 (Case 7)*		*0*	*0*	*1*	*CR post- therapy*
*Specimen 12 (Case 8)*		0	0	0	*CR post relapse post-alloSCT*
*Specimen 13 (Case 9)*		*2*	*1*	*2*	*AML MRD+ pre-alloSCT*
*Specimen 14 (Case 9)*		*0*	*0*	*0*	*CR post-alloSCT*
*Specimen 15 (Case 10)*		*0*	*0*	*0*	*Pre-alloSCT*
*Specimen 16 (Case 10)*		*0*	*0*	*3*	*CR post-alloSCT*
*Specimen 17 (Case 10)*		*2*	*1*	*3*	*Relapse 7 months post-alloSCT*
*Specimen 18 (Case 10)*		*2*	*2*	*3*	*Relapse 10 months post-alloSCT*
*Specimen 19 (Case 10)*		*3*	*3*	*4*	*Post therapy Azacitidine*
*Specimen 20 (Case 10)*		*3*	*3*	*3*	*Post therapy Azacitidine*
*Specimen 21 (Case 11)*		*5*	*5*	*6*	*Diagnosis of AML secondary to MDS*
*Specimen 22 (Case 11)*		*1*	*0*	*2*	*CR post-alloSCT*
*Specimen 23 (Case 11)*		*3*	*1*	*3*	*Relapse post-alloSCT*
*Specimen 24 (Case 12)*		*1*	*0*	*1*	*Pre-alloSCT*
*Specimen 25 (Case 12)*		*1*	*0*	*1*	*CR post-alloSCT*
*Specimen 26 (Case 12)*		*2*	*1*	*2*	*Relapse post-alloSCT*
*Specimen 27 (Case 13)*		*3*	*3*	*4*	*Relapse 2 months post-alloSCT*
*Specimen 28 (Case 13)*		*3*	*3*	*3*	*Relapse 4 months post-alloSCT under target therapy*
*Specimen 29 (Case 13)*		*2*	*1*	*2*	*Relapse 6 months post-alloSCT under target therapy*
*Specimen 30 (Case 14)*		*0*	*0*	*0*	*Pre-alloSCT*
*Specimen 31 (Case 14)*		*0*	*0*	*0*	*CR post-alloSCT*
*Specimen 32 (Healthy Control)*		*n.d.*	*0*	*0*	*Healthy status*
*Specimen 33 (Healthy Control)*		*n.d.*	*0*	*0*	*Healthy status*
*Specimen 34 (Healthy Control)*		*n.d.*	*0*	*0*	*Healthy status*

BM = Bone marrow cells DNA; PB = Peripheral blood cells DNA; Exo = DNA co-isolated with circulating exosomes; n.d. = not determined.

**Table 3 cancers-14-04504-t003:** Mutations detected in bone marrow cells’ DNA, peripheral blood cells’ DNA and circulating exosomes’ DNA in all the considered time points.

	*BM Cells*	*PB Cells*	*Exo dsDNA*
*Case 1*			
*Specimen 1*	*DNMT3A (c.327dupG; p.Q110Afs*13)* *ASXL1 (c.1927delG; p.G645Vfs*57)* *ASXL1 (c.1927dupG; p.G646Wfs*10)* *RUNX1 (c.667_668insA; p.R223Qfs*10)*	*DNMT3A (c.327dupG; p.Q110Afs*13)* *ASXL1 (c.1927delG; p.G645Vfs*57)* *ASXL1 (c.1927dupG; p.G646Wfs*10)*	*DNMT3A (c.327dupG; p.Q110Afs*13)* *ASXL1 (c.1927delG; p.G645Vfs*57)* *ASXL1 (c.1927dupG; p.G646Wfs*10)* *RUNX1 (c.667_668insA; p.R223Qfs*10)*
** *Case 2* **			
*Specimen 2*	*DNMT3A (c.G2189A; p.R730H)* *FLT3 (c.G2503T; p.D835Y)* *RUNX1 (c.337dupT; p.Y113Lfs*3)*	*DNMT3A (c.G2189A; p.R730H)* *FLT3 (c.G2503T; p.D835Y)*	*DNMT3A (c.G2189A; p.R730H)* *FLT3 (c.G2503T; p.D835Y)* *RUNX1 (c.337dupT; p.Y113Lfs*3)*
** *Case 3* **			
*Specimen 3*	n.d.	n.d.	n.d.
*Specimen 4*	*ETV6 (c.C1198G; p.H400D)*	n.d.	*ETV6 (c.C1198G; p.H400D)*
** *Case 4* **			
*Specimen 5*	*EZH2 (c.G1522A; p.G508R)* *ASXL1 (c.G4265C; p.S1422T)* *RUNX1 (c.G364C; p.A122P)*	*EZH2 (c.G1522A; p.G508R)* *ASXL1 (c.G4265C; p.S1422T)* *RUNX1 (c.G364C; p.A122P)*	*EZH2 (c.G1522A; p.G508R)* *ASXL1 (c.G4265C; p.S1422T)* *RUNX1 (c.G364C; p.A122P)* *TP53 (c.T194G; p.V65G)*
** *Case 5* **			
*Specimen 6*	*TET2 (c.A1532C; p.H511P)* *BCOR (c.A1589T; p.K530M)* *TP53 (c.C472A; p.R158S)*	*TET2 (c.A1532C; p.H511P)* *BCOR (c.A1589T; p.K530M)* *TP53 (c.C472A; p.R158S)*	*EZH2 (c.T748G; p.C250G)* *TET2 (c.A1532C; p.H511P)* *BCOR (c.A1589T; p.K530M)* *TP53 (c.C472A; p.R158S)*
*Specimen 7*	*EZH2 (c.T748G; p.C250G)* *TET2 (c.A1532C; p.H511P)* *BCOR (c.A1589T; p.K530M)* *TP53 (c C472A; p.R158S)* *TP53 (c.A1T; p.E2_M40del)*	*EZH2 (c.T748G; p.C250G)* *TET2 (c.A1532C; p.H511P)* *BCOR (c.A1589T; p.K530M)* *TP53 (c. C472A; p.R158S)*	*EZH2(c.T748G; p.C250G)* *TET2 (c.A1532C; p.H511P)* *BCOR (c.A1589T; p.K530M)* *TP53(c. C472A; p.R158S)*
*Specimen 8*	*TET2 (c.A1532C; p.H511P)*	n.d.	*TET2 (c.A1532C; p.H511P)*
** *Case 6* **			
*Specimen 9*	*TET2* (*c.*C4144A; p.H1382Y) *SF3B1* (*c.*A856T; p.I286F) *TP53* (*c.*T262A; p.S88T) *RUNX1* (*c.*G364C; p.A122P)	*TET2* (*c.* C4144A; p.H1382Y) *SF3B1* (*c.*A856T; p.I286F) *TP53* (*c.*T262A; p.S88T)	*TET2* (*c.*C4144A; p.H1382Y)
** *Case 7* **			
*Specimen 10*	*SRP72 (c.A926C; p.E309A)* *ETV6 (:c.G1167C; p.M389I)* *ASXL1 (c.C1260T; p.A420A)*	*ETV6 (:c.G1167C; p.M389I)* *ASXL1 (c.C1260T; p.A420A)*	*SRP72 (c.A926C; p.E309A)* *ETV6 (:c.G1167C; p.M389I)* *ASXL1 (c.C1260T; p.A420A)*
*Specimen 11*	n.d.	n.d.	*ASXL1 (c.C1260T; p.A420A)*
** *Case 8* **			
*Specimen 12*	n.d.	n.d.	n.d.
** *Case 9* **			
*Specimen 13*	*BCOR (c.G1306A; p.V436I)* *TP53 (c.C472A; p.R158S)*	*BCOR (c.G1306A; p.V436I)*	*BCOR (c.G1306A; p.V436I)* *TP53 (c.C472A; p.R158S)*
*Specimen 14*	n.d.	n.d.	n.d.
** *Case 10* **			
*Specimen 15*	n.d.	n.d.	n.d.
*Specimen 16*	n.d.	n.d.	*TET2* (c.C4144T; p.H1382Y) *ASXL1* (c.A4501T; p.S1501C) *RUNX1* (c.G1183T; p.E395X)
*Specimen 17*	*ASXL1* (c.A4501T; p.S1501C) *RUNX1* (c.G1183T; p.E395X)	*ASXL1* (c.A4501T; p.S1501C)	*ASXL1* (c.A4501T; p.S1501C) *RUNX1* (c.G1183T; p.E395X) *TET2* (c.C4144T; p.H1382Y)
*Specimen 18*	*ASXL1* (c.A4501T; p.S1501C) *RUNX1* (c.G1183T; p.E395X)	*ASXL1* (c.A4501T; p.S1501C) *RUNX1* (c.G1183T; p.E395X)	*ASXL1* (c.A4501T; p.S1501C) *RUNX1* (c.G1183T; p.E395X) *TET2* (c.C4144T; p.H1382Y)
*Specimen 19*	*TET2* (c.C4144T; p.H1382Y) *ASXL1* (c.A4501T; p.S1501C) *RUNX1* (c.G1183T; p.E395X)	*TET2* (c.C4144T; p.H1382Y) *ASXL1* (c.A4501T; p.S1501C) *RUNX1* (c.G1183T; p.E395X)	*TET2*(c.C4144T; p.H1382Y) *ASXL1* (c.A4501T; p.S1501C) *RUNX1* (c.G1183T; p.E395X) *IDH1* (c.A643T:p.I215F)
*Specimen 20*	*TET2* (c.C4144T; p.H1382Y) *ASXL1* (c.A4501T; p.S1501C) *RUNX1* (c.G1183T; p.E395X)	*TET2* (c.C4144T; p.H1382Y) *ASXL1* (c.A4501T; p.S1501C) *RUNX1* (c.G1183T; p.E395X)	*TET2* (c.C4144T; p.H1382Y) *ASXL1* (c.A4501T; p.S1501C) *RUNX1* (c.G1183T; p.E395X)
** *Case 11* **			
*Specimen 21*	*ASXL1* (c.1786dupG; p.R596P) *ASXL1* (c.A2957G; p.N986S) *EZH2 (c.T748G; p.C250G)* *IDH2 (c.435dupG; p.T146Dfs172)* *U2AF1 (c.A476G; p.E159G)*	*ASXL1* (c.1786dupG; p.R596P) *ASXL1* (c.A2957G; p.N986S) *EZH2 (c.T748G; p.C250G)* *IDH2 (c.435dupG; p.T146Dfs)* *U2AF1 (c.A476G; p.E159G)*	*ASXL1* (c.1786dupG; p.R596P) *ASXL1* (c.A2957G; p.N986S) *EZH2(c.T748G; p.C250)* *IDH2 (c.435dupG; p.T146Dfs)* *U2AF1 (c.A476G; p.E159G)* *TP53 (c.C523T; p.R175C)*
*Specimen 22*	*EZH2 (c.T748G; p.C250G)*	n.d.	*EZH2 (c.T748G; p.C250G)* *TP53 (c.C523T; p.R175C)*
*Specimen 23*	*EZH2 (c.T748G; p.C250G)* *ASXL1 (c.*A2957G; p.N986S) *TP53 (c.C523T; p.R175C)*	*EZH2 (c.T748G; p.C250G)*	*EZH2 (c.T748G; p.C250G)* *ASXL1 (c.*A2957G; p.N986S) *TP53 (c.C523T; p.R175C)*
** *Case 12* **			
*Specimen 24*	*ETV6 (c.*G1167C; p.M389I)	n.d.	*ETV6 (c.*G1167C; p.M389I)
*Specimen 25*	*ETV6 (c.*G1167C; p.M389I)	n.d.	*ETV6 (c.*G1167C; p.M389I)
*Specimen 26*	*ETV6 (c.*G1167C; p.M389I) *CEBPalpha* (c.564_566del; p.P189del)	*ETV6 (c.*G1167C; p.M389I)	*ETV6 (c.*G1167C; p.M389I) *CEBPalpha* (c.564_566del; p.P189del)
** *Case 13* **			
*Specimen 27*	*SRSF2 (c. 287dupC; p. P97Gfs27)* *KRAS (c.A9T; p. E3D)** *DNMT3A (c.G2189A; p.R730H)*	*SRSF2 (c. 287dupC; p. P97Gfs27)* *KRAS (c.A9T; p. E3D)** *DNMT3A (c.G2189A; p.R730H)*	*SRSF2 (c. 287dupC; p. P97Gfs27)* *KRAS (c.A9T; p. E3D)** *DNMT3A (c.G2189A; p.R730H)* *BCOR (c.A1589T; p.K530M)*
*Specimen 28*	*KRAS (c.A9T; p. E3D)** *DNMT3A (c.G2189A; p.R730H)* *BCOR (c.A1589T; p.K530M)*	*KRAS (c.A9T; p. E3D)** *DNMT3A (c.G2189A; p.R730H)* *SRSF2 (c. 287dupC; p. P97Gfs27)*	*KRAS (c.A9T; p. E3D)** *DNMT3A (c.G2189A; p.R730H)* *BCOR (c.A1589T; p.K530M)*
*Specimen 29*	*KRAS (c.A9T; p. E3D)** *BCOR (c.A1589T; p.K530M)*	*KRAS (c.A9T; p. E3D)**	*KRAS (c.A9T; p. E3D)** *BCOR (c.A1589T; p.K530M)*
** *Case 14* **			
*Specimen 30*	n.d.	n.d.	n.d.
*Specimen 31*	n.d.	n.d.	n.d.
** *Healthy Subject 1* **			
*Specimen 32*	n.a.	n.d.	n.d.
** *Healthy Subject 2* **			
*Specimen 33*	n.a.	n.d.	n.d.
** *Healthy Subject 3* **			
*Specimen 34*	n.a.	n.d.	n.d.

BM = Bone marrow cells’ DNA; PB = Peripheral blood cells’ DNA; Exo = circulating exosomes’ DNA; * = mutations confirmed by second target resequencing; n.d. = Not detected; n.a. = Not available.

## Data Availability

Data are available by Corresponding Author upon request.

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
