# Peer review of "Feasibility of Leukemia-Derived Exosome Enrichment and Co-isolated dsDNA Sequencing in Acute Myeloid Leukemia Patients: A Proof of Concept for New Leukemia Biomarkers Detection"

_cancers, 2022, doi:10.3390/cancers14184504_

Round 1

Reviewer 1 Report (New Reviewer)

The authors performed a pilot feasibility study on 14 cases with AML to determine the value of mutation analyses using DNA derived from leukemia derived exosomes including DNA surrounding the lipid membrane. From their data the authors conlcude that their approach proved feasible for  NGS analysis for AML biomarkers detection.

The study design is appropriate, however, the sample size quite small. More importantly in AML also fusion transcripts are important biomarkers, but these were neither analyzed nor even mentioned. Introducing a new technology into diagnostics requires a thoughtful analysis of pros and cons. Thus a table or a figure should be added providing this information comparing analysis of bone marrow cells, peripheral blood cells and exosome derived DNA. Further the authors should mention why they decided to analysis exosome derived DNA and not cell-free DNA.

It would be helpful if the authors could provide more details on the discrepancies between bone marrow, peripheral blood and exosomes, especially data on VAF would help to understand why some mutations might not have been detected in the other material. This data could be added in table 4.

Table 1 + Table 2: it would be helpful to combine both tables into one table or alternatively at least the case number of table 1 should be added to table 2 so that both tables can be related

As preanalytics are critical more details should be provided on the tubes used and the time between drawing peripheral blood or bone marrow until centrifugation (median and range) and whether or not the time variation had an impact on amount and quality of DNA.

Please specify: „leukemia burden estimated by cytofluorimetric analysis“ – how is leukemia burden defined? Only blast or all cells carrying an abeerant immunophenotype?

In addition to the correlation exosomal DNA concentration and leukemia burden, a correlation between the VAF of mutation derived from exosomal DNA vs bone marrow cell derived DNA and peripheral blood DNA would be of interest.

Lines 63: please change „myelodisplastic“ into „myelodysplastic“

Lines 248/249: „At least one mutation has been detected on exosomal dsDNA in 6 out of 10 (10%) specimens sam- 249 pled at the moment of Complete Remission (CR),…“please check numbers 6 out of 10 or 10%?

The format of table 4 should be optimized so that the same gene mutations are in the same line and if not detected in any of the material analyzed „n. d.“ für not detected are soemthing comparable should be inserted.

Author Response

We really thank the Reviewer for the support in the improvement of the present Manuscript. Please, find below the point by point review report.

The authors performed a pilot feasibility study on 14 cases with AML to determine the value of mutation analyses using DNA derived from leukemia derived exosomes including DNA surrounding the lipid membrane. From their data the authors conlcude that their approach proved feasible for NGS analysis for AML biomarkers detection.

  1. The study design is appropriate, however, the sample size quite small. More importantly in AML also fusion transcripts are important biomarkers, but these were neither analyzed nor even mentioned. Introducing a new technology into diagnostics requires a thoughtful analysis of pros and cons. Thus a table or a figure should be added providing this information comparing analysis of bone marrow cells, peripheral blood cells and exosome derived DNA. Further the authors should mention why they decided to analysis exosome derived DNA and not cell-free DNA.

We really thank the Reviewer for this point. The sample size is due to the characteristic of the study which is, at the moment, a proof of principle. Additional investigations on enlarged cohorts of patients are needed to confirm the reported results. The aim of the present study was the investigation of the feasibility of the detection of leukemic markers shuttled by or co-isolated with EVs released by leukemic cells. Considering the low abundancy of exosomal DNA when compared with exosomal RNA, we decided to face the most challenging approach. The feasibility of the strategy reported in the manuscript, will easily open to the analysis of the exosomal RNA, maybe by a contemporary analysis of RNA and DNA as suggested by recently developed NGS panel based on the analysis of fusion gene transcripts (RNA) and SNVs and indels (DNA). About cell free DNA, we have to consider that the vesicles isolation kit allows the co-precipitation of the surrounding DNA and, most importantly, moving from cell-free DNA to EVs will open to the simultaneous investigation of multiple targets/markers (e.g. miRNA, fusion transcripts, point mutations on DNA, active proteins). This seems to be more interesting in order to improve the diagnosis, the prognostication and the monitoring of complex diseases such as AML. Following reviewers suggestion, we added a sentence about the chromosomal rearrangements in the Introduction section, in the paragraph 1.2 Acute Myeloid Leukemias (AML)

  1. It would be helpful if the authors could provide more details on the discrepancies between bone marrow, peripheral blood and exosomes, especially data on VAF would help to understand why some mutations might not have been detected in the other material. This data could be added in table 4.

We really thank the reviewer about this point. The discrepancies between bone marrow and PB are known in literature (PMID: 32191348, 36030398, 34440157, 33851435, 32175599) and may rely on the presence of sub-clones resident in the BM and not yet circulating in the PB. On the other hand, PB samples may be enriched by clones with low homing signal and represent a good matrix for MRD monitoring. For this reason, we are trying to explore small vesicles as potential carriers of markers deriving from both BM and PB. Concerning the technical question, the Next Generation Sequencing (NGS) is considered a semi-quantitative technique. When compared with RT-qPCR or dPCR, NGS results defective in terms of robustness and reliability of quantification. For this reason, NGS applied to diagnosis limits its detection to 4% in term of quantification. Some of the mutations results with low VAF (<5%). If reviewer agrees, we prefer not to force the analysis which may results as speculation. Further analysis performed by dPCR or RT-qPCR will be assessed in order to investigate this aspect on enlarged cohort of patients/samples.

  1. Table 1 + Table 2: it would be helpful to combine both tables into one table or alternatively at least the case number of table 1 should be added to table 2 so that both tables can be related

Thank you for this suggestion. Table 1 and Table 2 have been combined (now Table 1). Please, see Materials and Methods section.

  1. As preanalytics are critical more details should be provided on the tubes used and the time between drawing peripheral blood or bone marrow until centrifugation (median and range) and whether or not the time variation had an impact on amount and quality of DNA.

Thank you for this point. PB and BM samples were not frozen. Samples were treated within 4 hours from sampling. Details concerning this preclinical aspects and the type of tube have been added from line 132 to line 135

  1. Please specify: „leukemia burden estimated by cytofluorimetric analysis“ – how is leukemia burden defined? Only blast or all cells carrying an abeerant immunophenotype?

Thank you for this question. Leukemia burden was defined on cells carrying aberrant IP. This detail has been added in the text.

  1. In addition to the correlation exosomal DNA concentration and leukemia burden, a correlation between the VAF of mutation derived from exosomal DNA vs bone marrow cell derived DNA and peripheral blood DNA would be of interest.

Thank you for this suggestion. It is an undiscussable interesting evaluation. As above mentioned, NGS is not recognized as a good technology for the quantification, while an optimal approach for detection of variants. In fact, many groups using NGS for MRD monitoring consider the “positivity” of the detection instead of the “quantity” of the disease.  We are not so confident in performing a correlation between the different relative VAF because of the quantification of low abundant variants. If Reviewer agrees, we could simply declare the different VAF by adding the information within Table 4.

  1. Lines 63: please change „myelodisplastic“ into „myelodysplastic“

We thank the reviewer for the identification of this typo error. The text has been edited accordingly.

  1. Lines 248/249: „At least one mutation has been detected on exosomal dsDNA in 6 out of 10 (10%) specimens sam- 249 pled at the moment of Complete Remission (CR),…“please check numbers 6 out of 10 or 10%?

We thank the reviewer for the identification of this typo error. The text has been edited accordingly.

  1. The format of table 4 should be optimized so that the same gene mutations are in the same line and if not detected in any of the material analyzed „n. d.“ für not detected are soemthing comparable should be inserted.

Thank you for this important suggestion. We edited the text changing “-“ in “n.d.” and moving the mutations in the same line. Please, see Table 3 (previously named Table 4).

Reviewer 2 Report (New Reviewer)

Bernardi et al.: Feasibility of leukemia-derived exosome...

In this paper the authors present the results of a pilot study which tested the use of NGS on dsDNA of exosomes in the conetext of AML. The authors found a linear relation between the tumor-burden and amount of exosomal DNA. Furthermore there was a 86,5% of homology between mutations found in the dsDNA and those found in the BM. This homology was diminished to 75% when the authors considered PB.

The paper is a very important piece of work as the linear relation between concentration of exosomal dsDNA and tumor burden could be used for a follow-up and an evaluation of the response to therapy.

The quality of the paper is flawed by some shortcomings in the text and lack of clarity.

Abstract. It is very clear and gives a good overview of the work done and the results obtained.

Major points:

1.) Introduction.

It is very hard to read the introduction as it is not subdivided in paragraphs. It needs a clear structure. It is a cut and paste exercise as the definitions and explanations are already there.

4.) Mat/Meth

  • A better explanation is needed in particular what the median age of 48years means, or the number of patients with WT1 over-expression means  in terms of, e.g., bias or other problems.

  • Is it possible to render Table1 and 2 clearer, maybe by combining them?

  • Other points have to be shortened; Blood sampling, plasma preparation, Ficoll or buffy coat preparation are widely established and routinely performed techniques. The same applies to cellular DNA extraction.

  • What does WGA mean for the overall result? Is there a possible bias, because of preferential amplification of some part of the DNA and not others?

  • The NGS part is limited: statements as "genes, known to be involved in AML", are a bit at least doubtful.

  • NGS: Please, explain the meaning of 1000x coverage.

  • NGS: How is it possible to detect "novel mutations" if the authors used a panel that covered only 27 genes?

  • The softwares have to be a bit better explained (e.g., Sophia DDM). What does it mean based on AI?

  • Statistical analysis: what are "standard descriptive statistics"? what is prism7.0 (brand etc...).

5.) Results.

  • cytofluorometric analysis - Mat/Meth?

  • linear regression: is there a threshold for the evaluation?

  • Table 3: combination with Table 1 and 2? for better clarity?

  • The authors should present better Table 4. For a reader it is much too long.

6.) Discussion

It is necessary that the authors shorten the discussion. They can eliminate all the repetition of their results and put them directly int the context of the literature.

Also a better structure by the introduction of more paragraphs would help.

6.) Discussion

It is necessary that the authors shorten the discussion. They can eliminate all the repetition of their results and put them directly int the context of the literature.

A better structure by the introduction of more paragraphs would also help.

Author Response

We really thank the Reviewer for the support in the improvement of the quality of the present manuscript. Please, find below the point by point review report.

In this paper the authors present the results of a pilot study which tested the use of NGS on dsDNA of exosomes in the conetext of AML. The authors found a linear relation between the tumor-burden and amount of exosomal DNA. Furthermore there was a 86,5% of homology between mutations found in the dsDNA and those found in the BM. This homology was diminished to 75% when the authors considered PB.

The paper is a very important piece of work as the linear relation between concentration of exosomal dsDNA and tumor burden could be used for a follow-up and an evaluation of the response to therapy.

The quality of the paper is flawed by some shortcomings in the text and lack of clarity.

Abstract. It is very clear and gives a good overview of the work done and the results obtained.

Thank you for your comment.

Major points:

1.) Introduction.

It is very hard to read the introduction as it is not subdivided in paragraphs. It needs a clear structure. It is a cut and paste exercise as the definitions and explanations are already there.

The Introduction has been subdivided in 3 paragraphs:

1.1 Extracellular vesicles (EVs) and liquid biopsy

1.2 Acute Myeloid Leukemias (AML)

1.3 A proof of concept for the study of AML markers by EVs

Please, see Introduction section at pages 2 and 3.

2.) Mat/Meth

A better explanation is needed in particular what the median age of 48years means, or the number of patients with WT1 over-expression means in terms of, e.g., bias or other problems.

The age is conventionally expressed as median (range) when describing a cohort of patients. This is the reason we reported “48 years (range 30-77 years)” in the text. The detailed age at enrollment is reported in Table 1. The overexpression of WT1, as well as the presence of additional mutation identified in the routine investigations are reported in order to undertint the type of AML, but are not expected to affect the analysis. In particular the WT1 over-expression, because it might influence the RNA analysis, and not the DNA analysis we are describing.

Is it possible to render Table1 and 2 clearer, maybe by combining them?

Thank you for this suggestion. Table 1 and Table 2 have been combined (now Table 1). Please, see Materials and Methods section.

Other points have to be shortened; Blood sampling, plasma preparation, Ficoll or buffy coat preparation are widely established and routinely performed techniques. The same applies to cellular DNA extraction.

We thank the reviewer for this suggestion. Considering that Ficoll and buffy coat have been performed specifically for the research project, so we consider it is important to detail as much as possible the procedure. Concerning the DNA extraction, many commercial kits allowing manual extraction are now available and some of them may include modifications to the procedure when applied for research. This is the reason we reported the different protocols. If Reviewers and Editors agree, we will remove that part.

What does WGA mean for the overall result? Is there a possible bias, because of preferential amplification of some part of the DNA and not others?

Thank you so much for this point. WGA has been widely explored in the recent years, in particular after the introduction of the single cell analysis (PMID: 28572171). It is a well established approach and it has been demonstrated its robustness when compared with other approaches (PMID: 31628895 ). Different WGA commercial kit are now available, with different performance/coverage (PMID: 34433869). RepliG is known to not affect the analysis by introducing bias and all the genes included in the present analysis are well covered (PMID: 34433869).

The NGS part is limited: statements as "genes, known to be involved in AML", are a bit at least doubtful.

Thank you for this comment. We modified this sentence in “genes known to be involved in myeloid leukemogenesis and suggested to be investigated by the recent international guidelines and by experts in the field.” Lease, see Materials and methods section, page 6, paragraph 2.6. Next Generation Sequencing (NGS) Analysis.

NGS: Please, explain the meaning of 1000x coverage.

In the context of NGS, 1000x coverage means that every locus has been sequenced/analyzed by at least 1000 reads.

NGS: How is it possible to detect "novel mutations" if the authors used a panel that covered only 27 genes?

Thank you for this question. The gene panel used in this project cover all the coding and regulatory sequences of 27 genes known to be involved in myeloid leukemogenesis. Some mutations or variants affecting these genes are known and have been already described in literature or reported in Ensembl.org. Every analysis may reveal or recognize novel variants or mutations never described before by other Research group. If a variant is not registered in a database, it has to be considered a novel variant (or mutation).

The softwares have to be a bit better explained (e.g., Sophia DDM). What does it mean based on AI?

We thank the reviewer for this question. Sophia DDM is a bioinformatic cloud based platform certified (CE-IVD) for NGS analysis in diagnostic workflow. It automatically queries a wide ranges of databases (e.g., such as OMIM, ClinVar, GnomAD, G1000, SIFT, MutationTaster) to reach a comprehensive variant annotation is based on different algorithms. Moreover, it is AI-based in terms of machine learning and ACMG-based pathogenicity ranking to classify variants confidently. Sophia DDM  analysis performs an algorithm-supported variant classification with OncoPortal™ to obtain the latest scientific evidence on all the relevant variants (machine-learning-based analysis). We add a short explanation in Materials and methods section, page 6, paragraph 2.6. Next Generation Sequencing (NGS) Analysis.

Statistical analysis: what are "standard descriptive statistics"? what is prism7.0 (brand etc...).

Thank you for this point. Descriptive statistics are an important part of biomedical research which is used to describe the basic features of the data in the study. e.g. the frequency of gene mutations, type of mutations and functions of mutated genes and the comparison of their frequency in the different samples’ matrix. We add a reference in the 2.7 Statistical analysis paragraph. Moreover, we better described the software used for the statistical analysis. It is Prism (GraphPad version 7.0). Now, Prism is distributed by Dotmatics. The present available version is 9.0.

3.) Results.

cytofluorometric analysis - Mat/Meth?

Cytofluorometric analysis were not conducted for the study. They were routinely performed by the diagnostic lab when required by clinicians following the clinical practice. No additional diagnostic exam was required for enrolled patients. Please see Page 3 and reference 7 and 26.

linear regression: is there a threshold for the evaluation?

No, linear regression analysis was performed on all the available data about leukemia burden and no threshold has been considered.

Table 3: combination with Table 1 and 2? for better clarity?

We thank the reviewer for this suggestion. However, Table 3 (now Table 2) reports some results data, while Table 1 and 2 (now merged, as Reviewer suggested) report data concerning the analyzed samples. The readability of the information reported in the Tables will results very hard when moving the type of sample in the Results section or when anticipating the results in the Materials and Methods. If Reviewers agrees, we think that Table 3 should remain in Results section.

The authors should present better Table 4. For a reader it is much too long.

Table 3 (previously named Table 4) has been edited. “-“ have been substituted with “n.d.” and mutations have been better aligned in the same line. We could require a special editing format for this table for a graphical form supporting the readability of the data.

4.) Discussion

It is necessary that the authors shorten the discussion. They can eliminate all the repetition of their results and put them directly int the context of the literature. Also a better structure by the introduction of more paragraphs would help.

Thank you so much for this suggestion. We has shorted the Discussion section.

This manuscript is a resubmission of an earlier submission. The following is a list of the peer review reports and author responses from that submission.

Round 1

Reviewer 1 Report

The revised manuscript by Bernardi et al is improved and several earlier comments have been addressed. For example, the authors increased the number of AML samples they analyzed from 10 to 14. While this number of patient samples is still too low to conclude whether or not sequencing of exosomal DNA will be the best method to detect biomarkers for leukemia, it does seem to show proof-of-principle that the method can at least be applied to categorize mutation status of leukemia cells. The authors describe at least one patient with a KRAS E3D mutation. I do not recall seeing reports of this mutation in the literature and wonder if the authors have identified a novel mutation or if this is some artefact of the procedure. This should be addressed and the authors should clearly state the weaknesses of their study.

Author Response

Reviewer 1

The revised manuscript by Bernardi et al is improved and several earlier comments have been addressed. For example, the authors increased the number of AML samples they analyzed from 10 to 14. While this number of patient samples is still too low to conclude whether or not sequencing of exosomal DNA will be the best method to detect biomarkers for leukemia, it does seem to show proof-of-principle that the method can at least be applied to categorize mutation status of leukemia cells. The authors describe at least one patient with a KRAS E3D mutation. I do not recall seeing reports of this mutation in the literature and wonder if the authors have identified a novel mutation or if this is some artefact of the procedure. This should be addressed and the authors should clearly state the weaknesses of their study.

We really thank the Reviewer for this point. We investigated the presence of KRAS c.A9T; p. E3D by resequencing the samples using Sophia Genetic Myeloid Solution and performing the bioinformatic analysis by Sophia DDM. Sophia Genetic is a certified solution for NGS analysis in the diagnostic workflow of hematological malignancies. The new analysis confirmed the presence of the mutation. Please, find below the screenshot of the report of Sophia DDM (The screenshot is in the attached file).

The additional analysis has been reported in Materials and Method and in Results section.

Reviewer 2 Report

Authors investigated the feasibility of a leukemia-derived exosomes enrichment approach in adult AML patients at different disease stages and evaluated the different types of molecular leukemic markers detectable in the studied exosomes fraction. In the current version I am forced to reject the manuscript. In particular,

  • In the manuscript absent evidence, that authors really work with exosomes. No data of transmission electron or kryo-electron microscopy, no data of NTA to confirm size of exosomes, absence of disrupted vesicles and etc…, no data of flow cytometry or western blotting to confirm exosomal markers such as CD9, CD81, etc…
  • Indirect confirmation that the samples obtained by the authors contain not only exosomes is the high concentration of exsomal DNA. Circulating DNA is a superposition of extracellular DNA and DNA from apoptotic bodies, various vesicles, including exosomes. Taking into account the extremely high concentrations of exosomal DNA found by the authors, corresponding to circulating DNA, I finally come to the conclusion that the authors have done a very large, complex and expensive job. However, what exactly they analyzed remained unclear .... and what information to extract from this work is also not clear ...

Author Response

Reviewer 2

Authors investigated the feasibility of a leukemia-derived exosomes enrichment approach in adult AML patients at different disease stages and evaluated the different types of molecular leukemic markers detectable in the studied exosomes fraction. In the current version I am forced to reject the manuscript. In particular,

In the manuscript absent evidence, that authors really work with exosomes. No data of transmission electron or kryo-electron microscopy, no data of NTA to confirm size of exosomes, absence of disrupted vesicles and etc…, no data of flow cytometry or western blotting to confirm exosomal markers such as CD9, CD81, etc…

Indirect confirmation that the samples obtained by the authors contain not only exosomes is the high concentration of exsomal DNA. Circulating DNA is a superposition of extracellular DNA and DNA from apoptotic bodies, various vesicles, including exosomes. Taking into account the extremely high concentrations of exosomal DNA found by the authors, corresponding to circulating DNA, I finally come to the conclusion that the authors have done a very large, complex and expensive job. However, what exactly they analyzed remained unclear .... and what information to extract from this work is also not clear ...

Thank you for this comment. Considering the aim of the project, the Authors selected a certified kit (CE-IVD certification is reported in the manuscript) to avoid the characterization of the isolated vesiscles. The manufacturer and the certifications ensure the users about the type and the origin of the DNA. This aspect has been discussed from line 322 to 334. In case of different investigation, such as the characterization of the isolated fraction, the kit is not the ideal approach. The Authors aimed to face the capability of exosomes to carry detectable leukemic molecular markers. We tested the combination of the cell of origin-based enrichment of vesiscles and the NGS in order to improve the sensitivity of the approach in a user-friendly workflow, suitable for further clinical application. The choice of a certified CE-IVD was made in this scenario during the experimental design, because clinical applications need certified kit and approaches and additional time-consuming validation tests would be useless and limiting. The certifications guarantee the quality.

Reviewer 3 Report

The authors have adequately addressed my major comment. A couple minor comments are listed below.

Minor

  1. Line 112: the age range for the patient samples needs to be adjusted to 30-77.
  2. Table 3: Since BM cells were not analyzed for the healthy controls, instead of “0” mutations detected in the BM it should be listed as not determined.

Author Response

Reviewer 3

The authors have adequately addressed my major comment. A couple minor comments are listed below.

Minor

Line 112: the age range for the patient samples needs to be adjusted to 30-77.

Table 3: Since BM cells were not analyzed for the healthy controls, instead of “0” mutations detected in the BM it should be listed as not determined.

Thank you so much for this suggestions. The age of the enrolled patients as well as Table 3 have been corrected following Reviewer’s comments.

Reviewer 4 Report

the manuscript needs English and typo errors correction.

Author Response

Reviewer 4

The manuscript needs English and typo errors correction.

Thank you for the revision. The manuscript has been revised both for English and for typo errors.

Round 2

Reviewer 2 Report

In the revised version of the manuscript, the authors did not consider it necessary to characterize the object they are working with (TEM, NTA, flow cytomenry or WB, etc.). The lack of confirmation of isolation of exosomes without impurities and only with references to kit manufacturer's certificates makes this work dubious! I carefully read all the documentation for this isolation kit: The manufacturer guarantees exosome isolation, but there is no word about co-isolation of cell-free DNA. Judging by the DNA concentrations that the authors find in exosome samples, this clearly indicates that the samples contain predominantly extracellular DNA. By the way, in an article recently published in mdpi (https://doi.org/10.3390/diagnostics12040854), an assessment was made of the proportion of exosomal DNA from the content of total extracellular DNA - exosomal DNA is only 0.03% of cell-free DNA! This finally convinced me to reject this manuscript.

Author Response

Q. In the revised version of the manuscript, the authors did not consider it necessary to characterize the object they are working with (TEM, NTA, flow cytomenry or WB, etc.). The lack of confirmation of isolation of exosomes without impurities and only with references to kit manufacturer's certificates makes this work dubious! I carefully read all the documentation for this isolation kit: The manufacturer guarantees exosome isolation, but there is no word about co-isolation of cell-free DNA. Judging by the DNA concentrations that the authors find in exosome samples, this clearly indicates that the samples contain predominantly extracellular DNA. By the way, in an article recently published in mdpi (https://doi.org/10.3390/diagnostics12040854), an assessment was made of the proportion of exosomal DNA from the content of total extracellular DNA - exosomal DNA is only 0.03% of cell-free DNA! This finally convinced me to reject this manuscript.

A. We thank the Reviewer for this comment. We edited the text following the indications provided by the Reviewer and the Editor and considered that the presented data derived from both the analysis of the exosomal cargoes and the surrounding DNA. All the text, the figures’ legends and the title have been revised following this suggestion.